# Association between Leisure-Time and Commute Physical Activity and Pre-Diabetes and Diabetes in the Brazilian Longitudinal Study of Adult Health (ELSA-Brasil)

**DOI:** 10.3390/ijerph20010806

**Published:** 2023-01-01

**Authors:** Yuri Sánchez-Martínez, Alessandra C. Goulart, Bianca de Almeida-Pititto, Bruce B. Duncan, Maria Inês Schmidt, Itamar de Souza Santos, Paulo A. Lotufo, William R. Tebar, Isabela M. Benseñor

**Affiliations:** 1Center for Clinical and Epidemiologic Research, University Hospital, University of São Paulo, São Paulo 05508-000, Brazil; 2Department of Preventive Medicine, Federal University of São Paulo, São Paulo 04023-062, Brazil; 3Postgraduate Program in Epidemiology, School of Medicine and Hospital de Clínicas, Federal University of Rio Grande do Sul, Porto Alegre 90035-003, Brazil

**Keywords:** lifestyle, exercise, chronic diseases, insulin resistance, epidemiology

## Abstract

Background: Diabetes is an important public health problem due to its health impairments and high costs for health services. We analyzed the relationship between the domains of physical activity at leisure-time (LTPA) and at commuting (CPA) with diabetes and pre-diabetes in an ELSA-Brasil study. Methods: Data from 11,797 participants (52.5% women, 49.1 ± 7.2 years) were analyzed. LTPA and CPA were measured using the International Physical Activity Questionnaire. Diabetes and pre-diabetes were defined by medical history, medication use to treat diabetes or blood glucose. Logistic regression models were performed to estimate the association between LTPA and CPA with diabetes and pre-diabetes after adjustment for sociodemographic and cardiovascular risk factors. Results: The prevalence of LTPA and CPA was 24.4% and 34%, respectively. Physically active participants at LTPA were less likely to have pre-diabetes (OR = 0.86 [95% CI = 0.77–0.95]) and diabetes (OR = 0.80 [95% CI = 0.69–0.93]), compared with inactive participants. No association between CPA and diabetes/pre-diabetes was observed. LTPA was inversely associated with diabetes among men (OR = 0.73 [95% CI = 0.60–0.89]), but was not associated among women. Women who were active (OR = 0.78 [95% CI = 0.67–0.90]) (OR = 0.79 [95% CI = 0.65–0.95]) at LTPA were less likely to have pre-diabetes, than inactive women. Conclusion: LTPA was inversely associated with diabetes and pre-diabetes in the ELSA-Brasil participants. A different behavior was observed between genders.

## 1. Introduction

The International Diabetes Federation estimated that the prevalence of diabetes worldwide in 2019 was 9.6% in men and 9.0% in women [1]. Additionally, the World Health Organization (WHO) reported an increase in diabetes prevalence from 4.7% to 8.5% between 1980 and 2014 [2]. Diabetes is considered an important public health problem due to its high and growing prevalence, its frequent complications and high costs for health services. Particularly in low and middle-income countries, this scenario is even worse because diabetes prevalence is growing faster than in developed countries and people have less access to health care [1,2].

Lifestyle characteristics, such as physical activity (PA) is a very important protective factor in prevention and treatment of diabetes [3]. Regular practice of PA improves heart and lung function and is associated with a lower risk of diabetes [4,5] and comorbidities such as cardiovascular disease [4]. Additionally, PA also helps in weight control, improves lipoprotein profile [4] and insulin sensitivity [4,5]. Particularly, in patients with diabetes, regular practice of PA might decrease glycated hemoglobin (HbA1c) and insulin resistance [5].

The impact of the domain where PA is practiced on diabetes is less studied [6,7]. PA can be measured in several life domains, such as household, at work, in commuting-time (CPA) and at leisure-time (LTPA) [4]. The association between PA and diabetes has been measured mostly in the LTPA domain [8,9] or considering the total amount from all domains together [10]. However, there are some conflicting results in the association of LTPA and CPA and diabetes. Data from previous studies showed that the magnitude of the association between PA and diabetes could be different, depending on the domain of PA evaluated [7,10,11]. A Japanese occupational cohort evidenced an inverse association between PA and diabetes incidence only at LTPA, but not at CPA [7]. An additional cross-sectional study found that both LTPA and CPA were associated with a lower prevalence of diabetes with a stronger association at CPA compared to LTPA [11]. Conversely, a meta-analysis of prospective studies supports that although PA is inversely associated with diabetes independent of the domain assessed, the strongest association occurred at the leisure-time domain [10]. 

Considering these conflicting findings about the relationship between PA and diabetes, the aim of this study was to cross-sectionally analyze the association of LTPA and CPA with the prevalence of diabetes and pre-diabetes in the Brazilian Longitudinal Study of Adult Health (ELSA-Brasil). We hypothesized that active status in the LTPA domain is more associated with lower diabetes/pre-diabetes prevalence than in the CPA domain.

## 2. Materials and Methods

ELSA-Brasil is a prospective cohort study of 15,105 civil servants aged 35–74 years in six different state capitals in Brazil with a focus on the incidence of cardiovascular disease and diabetes and associated factors. Further details on the study design and cohort profile can be found elsewhere [12,13,14]. This is a cross-sectional analysis using data from the baseline examination (2008–2010). After the exclusion of retired workers (n = 3009), participants reporting previous coronary heart disease (N = 532), stroke (n = 197) and with missing information about LTPA (n = 220), CPA (n = 245) and diabetes (n = 4), 11,797 participants were included in the analysis. 

Women less than four months after childbirth people with cognitive or communication impairment and those living outside the metropolitan area of the city were not eligible to participate. Participants had an initial interview using validated questionnaires and were scheduled for clinical examinations and laboratory tests in the research center. The study was conducted in accordance with the Declaration of Helsinki and was approved by the local institutional review boards. All participants signed the informed consent prior to enrollment.

### 2.1. Physical Activity Measurement

PA was estimated by applying the International Physical Activity Questionnaire (IPAQ, long version), in the domains of LTPA and CPA. The instrument was validated in Brazil [15]. The pattern of PA in its different domains was reported in min per week and calculated by multiplying the weekly rate for the duration of each activity. Issues related to the level of LTPA and CPA were classified according to the recommendations of the WHO. The categories created were: (1) physically active (≥150 min per week of moderate activity or ≥75 min a week of vigorous activity); (2) insufficiently active (<150 min per week of moderate activity and or <75 min a week of vigorous activity); and physically inactive when no PA was reported [16]. 

### 2.2. Diabetes Diagnosis

A 12-h fasting blood sample was drawn in the morning soon after arrival at the research center, following standardized procedures. A standardized 75 g oral glucose tolerance test (OGTT) was performed in all participants without known diabetes, utilizing an anhydrous glucose solution with plasma glucose levels measured after 2 h. Plasma glucose was measured by the hexokinase method (ADVIA Chemistry; Siemens, Deerfield, Illinois). HbA1C was measured by high-pressure liquid chromatography (Bio-Rad Laboratories, Hercules, California) using a method certified by the National Glycohemoglobin Standardization Program [17]. Meanwhile, diabetes status was defined in a comprehensive fashion as follows: a previously diagnosed diabetes was classified when answering “yes” to either “Have you been previously told by a physician that you had/have diabetes (sugar in the blood)?” or “Have you used medication for diabetes in the past 2 weeks?” Previously undiagnosed diabetes was classified based on laboratory values when reaching the threshold for fasting plasma glucose (FPG; ≥7.0 mmol/L), 2-h plasma glucose (2h-PG ≥11.1 mmol/L), or HbA1c (≥6.5%; ≥47.5 mmol/mol) [15]. Pre-diabetes was defined through fasting plasma glucose (FPG; ≥5.6 mmol/L to 6.9 mmol/L); or 2-h plasma glucose (2-h PG; ≥7.8 mmol/L to 11.0 mmol/L) or HbA1c (5.7–6.4%) [18]. 

### 2.3. Covariates

Sociodemographic and health-related variables such as age (years), educational attainment (less than high school, complete high school and incomplete college, and at least complete college), average family monthly income (less than US$ 1245, US$ 1245–3734 and at least US$ 3735; 1 US$ = R$ 2.00), marital status (married, not married), self-reported race (white, mixed, black, and others) and use of medications to treat diabetes were measured through a structured questionnaire. Macronutrients daily intake was calculated based on information from a food frequency questionnaire validated for the study [19]. Anthropometric measurements were performed by trained personnel, using standard equipment and techniques [20]. Body mass index (BMI) were calculated as weight by squared height, and categorized as underweight (<18.5 kg/m^2^), normal weight (18.5–24.9 kg/m^2^), overweight (25–29.9 kg/m^2^) and obesity (≥30 kg/m^2^); waist circumference was classified by cardiovascular risk, using as cut point 88 cm in women and 102 cm in men; hypertension was defined as systolic/diastolic blood pressure ≥140/90 mmHg or use of antihypertensive medications; dyslipidemia was defined as triglycerides ≥150 mg/dL or LDL-cholesterol ≥130 mg/dL or HDL-cholesterol <40 mg/dL in men and <50 mg/dL in women or taking lipid-lowering medication. Alcohol intake and smoking status were self-reported as never, past and current user. 

### 2.4. Statistical Analysis 

Categorical variables were presented as counts and proportions and compared using a Chi-square test. Continuous variables were presented as mean (standard deviations) and compared using one-way ANOVA, or median [interquartile range] and compared using the Kruskal–Wallis test, if necessary. We found an interaction between LTPA (*p* = 0.038) and CPA with sex (*p* = 0.019) in participants with pre-diabetes. Therefore, logistic models were presented according to sex. Logistic regression models were performed to estimate the association between LTPA/CPA as the independent variable and pre-diabetes/diabetes as the dependent variable. Odds Ratio (OR) and 95% Confidence Interval (95% CI) were presented as crude, adjusted by sociodemographic variables (age, education and race) and with multivariable adjustment for sociodemographic and cardiovascular risk factors (age, education, race, waist circumference, hypertension, smoking, alcohol intake and use of medications to treat diabetes). LTPA models were adjusted by CPA and CPA models were adjusted by LTPA. It was considered as significant *p*-values < 0.05. All analyses were developed with Stata Statistical Software: Release 15. College Station, TX: StataCorp LLC. 

## 3. Results

From the 11,797 individuals, 52.5% were women. Mean age was 48.8 (7.1) for women and 49.4 (7.4) for men. Overall, 53% had completed college or higher, 51% were white, 30.1% had hypertension and 46.0% dyslipidemia. Frequency of dyslipidemia was higher in participants with diabetes (51.5%), compared to people without diabetes (40.5%). Global, 50.1% reported pre-diabetes, 14.1% diabetes and 7.8% used at least one medication to treat diabetes. Among participants under pharmacological treatment, 90.4% used metformin, 18.3 glibenclamide, 10.3% glimepiride, 7.4% glicazide, 3% vildagliptin, 2.2% sitagliptin and 1.6% acarbose for diabetes management. Other medications used to diabetes were chlorpropamide, exenatide, pioglitazone, repaglinde, rosiglitazone and saxagliptin with frequencies of use less than 1%. Only 24.4% and 34% were considered physically active in the LTPA and CAP domains, respectively. 

Table 1 describes the sociodemographic and clinical characteristics of the sample according to level of LTPA by sex. Both active women and men at leisure-time were mostly white, with college attainment, never smokers and with lower waist circumference compare to other groups. Table 2 presents the same data according to CPA with similar findings.

Table 3 describes the logistic models for LTPA and CPA and diabetes for all samples, both men and women. For the entire sample, there was an association between LTPA and diabetes (OR = 0.80; 95% CI = 0.69–0.93); no association was found between CPA and diabetes. In men, there was an inverse association of LTPA and diabetes at the crude model (OR = 0.46; 95% CI = 0.39–0.56) that remained significant after multivariable adjustment (OR = 0.73; 95% CI = 0.60–0.89). For CPA, the crude and multivariable models did not show any association of practice of PA and diabetes among men. For women, there was a crude and inverse association of LTPA with diabetes (OR = 0.63; 95% CI = 0.50–0.78) that remained significant after adjusting for the sociodemographic factors (OR = 0.73; 95% CI = 0.58–0.92) but lost significance after multivariable adjustment (OR = 0.88; 95% CI = 0.69–1.12). No associations were found between CPA and diabetes in the multivariable models for women. 

Table 4 presents logistic models for LTPA and CPA and pre-diabetes for the entire sample, men and women. Overall, active persons have a lower prevalence of pre-diabetes compared with inactive ones (OR = 0.86, 95% CI = 0.77–0.95). In men, an inverse crude association between LTPA and pre-diabetes lost significance after further adjustment. For CPA, there was an inverse association with pre-diabetes after sociodemographic adjustment that lost significance after multivariable adjustment. For women, in the crude and multivariable models, LTPA was associated with a lower prevalence of pre-diabetes for insufficiently active (OR= 0.79; 95% CI= 0.65–0.95) and active (OR = 0.78; 95% CI= 0.67–0.90) as compared to inactive women. No association was found in women for CPA and pre-diabetes. This section may be divided by subheadings. It should provide a concise and precise description of the experimental results, their interpretation, as well as the experimental conclusions that can be drawn.

## 4. Discussion

Our results showed that male participants who were physically active or insufficiently active at the LTPA domain were less likely to have diabetes when compared to those who were inactive, while no association was observed between physical activity levels at the CPA domain and diabetes. For women, no significant association was found between LTPA or CPA and diabetes. Considering only pre-diabetes, no association was found for LTPA or CPA with pre-diabetes in men. In contrast, active and insufficiently active women at LTPA were less likely to have pre-diabetes than inactive women. Considering the entire sample, we found that participants in the physically active group were less likely to have pre-diabetes and diabetes when compared with inactive participants. Overall, LTPA was more associated with less diabetes and pre-diabetes compared to CPA, with a different behavior according to sex.

The present study findings for men were in agreement with the findings from the European Health Interview Survey in Spain [9], a nationwide survey that involved people aged 18–74 years. In the study, men who were physically active had a lower prevalence of diabetes compared to inactive men. In another cross-sectional study in Wales, regular PA was associated with lower diabetes prevalence among men [21]. Additionally, other cohort studies have identified an inverse association of being active in LTPA domain with diabetes in men [8,22]. However, with respect to pre-diabetes in men, our results did not show any significant association with PA domains. It agrees with the findings from a cross-sectional study in younger people. As we did, they defined physically active people as those who practice at least 150 min per week of PA. Finally, they reported no differences in abnormal glucose metabolism between physically active men compared with inactive ones [23]. Another cross-sectional study, using its own PA questionnaire, showed a borderline inverse association between PA and pre-diabetes in elderly men [24]. 

There are few studies about the association between CPA and diabetes by sex. Our results about CPA showed no association with the diabetes/pre-diabetes prevalence in men. This is not in agreement with the findings from a Chinese study developed by Hu et al. (2018) [6], who also used IPAQ showed an inverse association between CPA and altered glucose levels in men. Regarding to CPA and diabetes in women, similar to our findings, this Chinese cross-sectional study did not report a significant association between CPA and in women between 26–77 years of age. 

In women, we observed no association between LTPA and diabetes. Data about the association of LTPA and diabetes in women are not consistent in the literature [8,9]. Our findings are different from The European Health Interview Survey in Spain, which evidenced a significant association between LTPA and a lower prevalence of diabetes in women [9]. Additionally, two prospective cohort studies—the InterAct project, developed in eight European countries [8] and the Nurses’ Health Study in USA [25]—found that physically active women at leisure-time had a lower risk of developing diabetes compared with inactive women. Regarding pre-diabetes in women, we found an inverse association with LTPA and two cross-sectional studies are consistent with our results. The AusDiab Study reported that 150 min per week of LTPA was inversely associated with abnormal glucose metabolism in women over 25 years [23]. Besides, the Elderly Nutrition and Health Survey in Taiwan found that women in the 2nd tertile of PA had a lower chance of impaired fasting glycemia compared with those in the 1st tertile [24]. 

These conflicting results, especially for women, may be due to differences in the design of the analysis, such as prospective [8,25] or cross-sectional [6,9] and the age-strata of participants, which is extremely variable with the inclusion of young adults in previous studies [6,9], compared to the ELSA-Brasil participants. Moreover, LTPA and CPA were reported in different ways among studies, as well as diabetes, which was self-reported in some studies [8,9] compared to ours, which used a very comprehensive approach. These methodological differences among studies can be partially responsible for the heterogeneity observed in the results. In addition, the sample size in other studies is larger than in ELSA-Brasil [8,9]. It is important to note that the heterogeneity in the results is higher in studies in women compared to men.

When we analyzed men and women together, our results showed an inverse association between LTPA and diabetes. These findings agree with the results from a Spanish cross-sectional study, which found that people who practiced at least 150 min weekly of LTPA had a significant lower prevalence of diabetes as compared to inactive people [9]. Similarly, Aune et al. (2015) [10] reported that active people at leisure time had a lower risk of diabetes compared with inactive ones. When we analyzed the pre-diabetes in the entire sample (men and women together), we found an inverse association with LTPA. We did not find any previous studies regarding LTPA and its association with pre-diabetes in women and men together to compare our results. Nevertheless, the results from a cross-sectional study, which included people between 35–60 years in Eastern Uganda showed that 150 min weekly of global PA was associated with a lower likelihood of abnormal glucose regulation [26]. Again, differences in the measurement of PA and in the age-strata included in the analyses may partially explain differences in the results among studies. 

The inverse association observed between PA and diabetes can be explained through weight control and improvements in the lipid profile. The chronic elevation of plasmatic lipids contributes to their accumulation in body tissues, such as skeletal muscle and liver and heart and pancreas; in turn, it can lead to insulin resistance [27,28,29]. There is evidence suggesting that the regular practice of PA may not only be associated with a better weight control but also with an enhancement in the lipid profile [4,30,31]; additionally, PA can contribute to decreased fat mass, particularly visceral fat [29], which is more related to insulin resistance than subcutaneous fat [28]. We hypothesized that the practice of LTPA was associated with less diabetes than CPA. This was true for men but not for women. Probably, LTPA has a greater relevance than CPA on health. LTPA is an activity that people develop as a personal decision, and most of the time, it is a pleasant activity. Moreover, it is possible that people reach more moderate and vigorous intensities during LTPA than CPA, and it has been previously reported that intensity is an important determinant of the physiological responses to PA [4]. 

Additionally, people who practice CPA could be more exposed to polluted environments than those who practice LTPA. Previously, it has been reported that during active transport, people are exposed to air pollution, specifically to particular matter 2.5 (PM2.5) or 10 (PM10) and nitrogen dioxide (NO_2_) [32]. Exposure to PM2.5 and PM10 has been associated to oxidative stress and systematic inflammation, which, in turn, can lead to insulin resistance and alter glucose metabolism [33]. A previous study found that individuals who use active transport everyday have an increased frequency of diabetes [34]. Similarly, another study reported increases in serum glucose and HbA1c and lipid markers associated to higher exposure levels to PM2.5 and PM10 [33].

This study has some limitations. The cross-sectional design does not allow to establish a causal association between exposures and outcomes. It also permits reverse causality as those with diabetes or pre-diabetes are counseled to do more PA, thus making it harder to find associations. Self-reported PA can introduce a measurement bias and some kind of misclassification is possible. As strengths, we have to highlight the large sample size and the standardized process to collect information under strict quality control. We had objective measures of diabetes including not only medical history and use of medication to treat diabetes but also fasting plasma glucose, 2-h plasma glucose, and HbA1C. PA was measured through a validated questionnaire used worldwide. In addition, we adjusted the analysis by several potential confounders.

## 5. Conclusions

LTPA is associated with a lower frequency of diabetes in men and a lower frequency of pre-diabetes in women, thus suggesting different behavior by sex. Considering both sexes together, the practice of LTPA was also associated with a lower prevalence of diabetes and pre-diabetes while CPA was neither associated with diabetes nor pre-diabetes, regardless of sex. Further analyses with prospective data from ELSA-Brasil to verify causal inference between PA and diabetes according to sex is guaranteed to better explore this relationship.

## Figures and Tables

**Table 1 ijerph-20-00806-t001:** Descriptive characteristics according to physical activity level at leisure-time in baseline participants of ELSA-Brasil study (n = 11,797).

	Women				Men				Overall Sample			
	Physically Inactive (n = 4179)	Insufficiently Active (n = 666)	Physically Active (n = 1253)	*p*-Value	Physically Inactive (n = 3182)	Insufficiently Active (n = 748)	Physically Active (n = 1579)	*p*-Value	Physically Inactive (n = 7361)	Insufficiently Active (n = 1414)	Physically Active (n = 2832)	*p*-Value
Age ^1^	48.8 ± 7.0	49.7 ± 7.1	48.6 ± 7.2	0.003	50.0 ± 7.4	49.6 ± 7.4	48.2 ± 7.2	0.001	49.3 ± 7.2	49.3 ± 7.2	48.4 ± 7.2	<0.001
Educational level, n (%)												
<High school	371 (8.9)	23 (3.4)	41 (3.3)	<0.001	583 (18.3)	110 (14.7)	128 (8.1)	<0.001	954 (13.0)	133 (9.4)	169 (6.0)	<0.001
High school	1812 (43.4)	141 (21.2)	294 (23.5)		1218 (38.3)	244 (32.6)	484 (30.6)		3030 (41.1)	385 (27.2)	778 (27.5)	
At least college	1996 (47.7)	502 (75.4)	918 (73.2)		1381 (43.4)	394 (52.7)	967 (61.2)		3377 (45.9)	896 (63.4)	1885 (66.5)	
Family income (US$), n (%)												
<1245	1323 (31.8)	110 (16.6)	195 (15.6)	<0.001	999 (31.5)	189 (25.4)	333 (21.1)	<0.001	2322 (31.7)	299 (21.2)	528 (18.7)	<0.001
1245–3734	2259 (54.3)	349 (52.6)	668 (53.4)		1546 (48.8)	357 (48.0)	768 (48.7)		3805 (51.9)	706 (50.1)	1436 (50.7)	
≥3735	577 (13.9)	205 (30.8)	388 (31.0)		621 (19.6)	198 (26.6)	477 (30.2)		1198 (16.3)	403 (28.6)	865 (30.6)	
Married, n (%)	2339 (56.0)	368 (55.3)	657 (52.4)	0.087	2599 (81.7)	609 (81.4)	1255 (79.5)	0.182	4938 (67.1)	977 (69.1)	1912 (67.5)	0.316
Race, n(%)												
Black	858 (20.7)	93 (14.1)	157 (12.6)	<0.001	471 (15.0)	104 (14.1)	200 (12.8)	0.001	1329 (18.2)	197 (14.1)	357 (12.8)	<0.001
Mixed	1215 (29.4)	143 (21.6)	303 (24.4)		1031 (32.8)	204 (27.7)	470 (30.2)		2246 (30.8)	347 (24.8)	773 (27.6)	
White	1940 (46.9)	393 (59.5)	716 (57.7)		1527 (48.6)	411 (55.8)	843 (54.2)		3467 (47.6)	804 (57.5)	1559 (55.7)	
Other	124 (3.0)	32 (4.8)	65 (5.2)		115 (3.6)	17 (2.3)	43 (2.8)		239 (3.3)	49 (3.5)	108 (3.9)	
BMI ^2^, n (%)												
Underweight	40 (1.0)	10 (1.5)	10 (0.8)	<0.001	40 (1.3)	8 (1.1)	11 (0.7)	<0.001	80 (1.1)	18 (1.3)	21 (0.7)	<0.001
Normal weight	1509 (36.1)	301 (45.2)	644 (51.4)		989 (31.1)	253 (33.9)	608 (38.5)		2498 (33.9)	554 (39.2)	1252 (44.2)	
Overweight	1495 (35.8)	220 (33.0)	412 (32.9)		1400 (44.0)	345 (46.2)	721 (45.7)		2895 (39.3)	565 (40.0)	1133 (40.0)	
Obese	1133 (27.1)	135 (20.3)	187 (14.9)		752 (23.6)	141 (18.8)	238 (15.1)		1885 (25.6)	276 (19.5)	425 (15.0)	
WC (CV risk) ^3^, n (%)	1912 (45.7)	247 (37.1)	395 (31.5)	<0.001	931 (29.3)	174 (23.3)	260 (16.5)	<0.001	2843 (38.6)	421 (30.0)	655 (23.1)	<0.001
Pre-diabetes, n (%)	1710 (44.0)	233 (36.9)	416 (34.5)	<0.001	1729 (63.1)	408 (62.2)	824 (56.1)	<0.001	3439 (51.9)	641 (49.8)	1240 (46.4)	<0.001
Diabetes mellius, n (%)	532 (12.7)	63 (9.5)	105 (8.38)	<0.001	649 (20.4)	132 (17.6)	168 (10.6)		1181 (16.0)	195 (13.8)	273 (9.6)	<0.001
Medications to treat diabetes, n(%)	198 (4.8)	34 (5.1)	46 (3.7)	0.217	259 (8.2)	51 (6.8)	65 (4.1)	<0.001	457 (6.2)	85 (6.0)	111 (3.9)	<0.001
Hypertension, n (%)	1185 (28.4)	147 (22.1)	262 (20.9)	<0.001	1231 (38.7)	263 (35.2)	415 (26.3)	<0.001	2416 (32.8)	410 (29.0)	677 (23.9)	<0.001
Dyslipidemia, n(%)	1696 (40.7)	275 (41.3)	484 (38.7)	0.399	1396 (44.0)	332 (44.4)	683 (43.5)	0.906	3092 (42.1)	607 (43.0)	1167 (41.4)	0.599
Daily carbohydrate intake (g) ^1^	309.9 ± 124.2	289.6 ± 107.0	284.2 ± 108.9	<0.001	379.4 ± 155.8	374.3 ± 157.9	382.2 ± 163.7	0.527	339.9 ±142.9	334.5 ± 142.7	338.9 ± 150.2	0.442
Daily fat intake (g) ^1^	82.4 ± 35.9	80.1 ± 32.4	76.9 ± 30.6	<0.001	101.1 ± 45.3	99.5 ± 40.9	103.2 ± 44.5	0.126	90.5 ± 41.3	90.4 ± 38.4	91.6 ± 41.1	0.447
Daily protein intake (g) ^1^	121.2 ± 52.2	121.3 ± 48.6	118.5 ± 46.4	0.242	144.2 ± 64.2	143.3 ± 60.7	151.2 ±64.1	0.001	131.1 ± 58.8	133.0 ± 56.4	136.7 ± 59.2	<0.001
Alcohol intake, n (%)												
Never	701 (16.8)	78 (11.7)	135 (10.8)	<0.001	142 (4.5)	27 (3.6)	62 (3.9)	<0.001	843 (11.5)	105 (7.4)	197 (6.9)	<0.001
Former user	901 (21.6)	107 (16.1)	181 (14.5)		693 (21.8)	135 (18.0)	237 (15.0)		1594 (21.7)	242 (17.1)	418 (14.8)	
Current user	2572 (61.6)	479 (72.1)	935 (74.7)		2344 (73.7)	586 (78.3)	1280 (81.1)		4916 (66.8)	1065 (75.4)	2215 (78.3)	
Smoking status, n (%)												
Never smoker	2592 (62.0)	424 (63.7)	850 (67.8)	<0.001	1525 (47.9)	407 (54.4)	984 (62.3)	<0.001	4117 (55.9)	831 (58.8)	1834 (64.8)	<0.001
Former smoker	990 (23.7)	167 (25.1)	307 (24.5)		1086 (34.1)	244 (32.6)	445 (28.2)		2076 (28.2)	411 (29.1)	752 (26.5)	
Current smoker	597 (14.3)	75 (11.2)	96 (7.7)		571 (17.9)	97 (13.0)	150 (9.5)		1168 (15.9)	172 (12.1)	246 (8.7)	

^1.^ Mean and standard deviation. ^2.^ BMI: Body mass index, categorized as underweight (<18.5 kg/m^2^), normal weight (18.5–24.9 kg/m^2^), overweight (25–29.9 kg/m^2^) and obesity (≥30 kg/m^2^). ^3.^ WC: Waist circumference, considered as CV (cardiovascular risk) as ≥88 cm in women and ≥102 cm in men.

**Table 2 ijerph-20-00806-t002:** Descriptive characteristics, according to level of physical activity at commuting in baseline participants of ELSA-Brasil study (n = 11,797).

	Women				Men				Overall Sample			
	Inactive (n = 1776)	Insufficiently Active (n = 2370)	Active (n = 1943)	*p*-Value	Inactive (n = 1415)	Insufficiently Active (n = 2092)	Active (n = 1998)	*p*-Value	Inactive (n = 3191)	Insuffi ciently Active (n = 4462)	Active (n = 3941)	*p*-Value
Age ^1^	48.3 ± 7.0	48.7 ± 7.2	49.5 ± 6.9	<0.001	48.9 ± 7.3	49.4 ± 7.4	49.8 ± 7.4	0.002	48.6 ± 7.1	49.1 ± 7.3	49.7 ± 7.2	<0.001
Educational level, n (%)												
<High school	81 (4.6)	191 (8.1)	160 (8.2)	<0.001	155 (10.9)	299 (14.3)	366 (18.3)	<0.001	236 (7.4)	490 (11.0)	526 (13.3)	<0.001
High school	515 (29.0)	834 (35.2)	895 (46.1)		422 (29.8)	665 (31.8)	857 (42.9)		937 (29.4)	1499 (33.6)	1752 (44.5)	
At least college	1180 (66.4)	1345 (56.7)	888 (45.7)		838 (59.2)	1128 (53.9)	775 (38.8)		2018 (63.2)	2473 (55.4)	1663 (42.2)	
Family income (US$), n (%)												
<1245	290 (16.4)	641 (27.1)	694 (35.9)	<0.001	263 (18.6)	520 (24.9)	736 (37.1)	<0.001	553 (17.4)	1161 (26.1)	1430 (36.5)	<0.001
1245–3734	983 (55.6)	1277 (54.0)	1012 (52.3)		711 (50.3)	1012 (48.5)	946 (47.7)		1694 (53.2)	2289 (51.4)	1958 (50.0)	
≥3735	496 (28.0)	446 (18.9)	227 (11.7)		439 (31.1)	555 (26.6)	302 (15.2)		935 (29.4)	1001 (22.5)	529 (13.5)	
Married, n (%)	1082 (60.9)	1337 (56.4)	942 (48.5)	<0.001	1159 (81.9)	1718 (82.1)	1583 (79.2)	0.038	2241 (70.2)	3055 (68.5)	2525 (64.1)	<0.001
Race, n(%)												
Black	210 (11.9)	446 (19.0)	449 (23.4)	<0.001	168 (12.0)	261 (12.6)	346 (17.5)	<0.001	378 (11.9)	707 (16.0)	795 (20.4)	<0.001
Mixed	463 (26.2)	663 (28.3)	533 (27.8)		418 (29.9)	645 (31.2)	641 (32.5)		881 (27.9)	1308 (29.7)	1174 (30.2)	
White	1031 (58.4)	1152 (49.1)	864 (45.0)		766 (54.8)	1099 (53.2)	914 (46.4)		1797 (56.8)	2251 (51.0)	1778 (45.7)	
Other	61 (3.5)	85 (3.6)	74 (3.8)		45 (3.2)	59 (2.9)	70 (3.5)		106 (3.3)	144 (3.3)	144 (3.7)	
BMI ^2^, n (%)												
Underweight	18 (1.0)	28 (1.2)	14 (0.7)	0.066	16 (1.1)	16 (0.8)	27 (1.3)	0.002	34 (1.1)	44 (1.0)	41 (1.0)	0.454
Normal weight	731 (41.2)	985 (41.5)	735 (37.8)		436 (30.8)	687 (32.8)	724 (36.3)		1167 (36.6)	1672 (37.5)	1459 (37.0)	
Overweight	594 (33.5)	822 (34.7)	707 (36.4)		638 (45.1)	947 (45.3)	880 (44.1)		1232 (38.6)	1769 (39.6)	1587 (40.3)	
Obese	431 (24.3)	535 (22.6)	487 (25.1)		325 (23.0)	441 (21.1)	365 (18.3)		756 (23.7)	976 (21.9)	852 (21.6)	
WC (CV risk) ^3^, n (%)	749 (42.2)	947 (40.0)	854 (43.9)	0.029	402 (28.4)	540 (25.8)	423 (21.2)	<0.001	1151 (36.1)	1487 (33.3)	1277 (32.4)	0.004
Pre-diabetes, n (%)	660 (39.0)	876 (39.5)	818 (45.2)	<0.001	771 (61.7)	1134 (61.1)	1054 (60.1)	0.676	1431 (48.6)	2010 (49.3)	1872 (52.6)	0.002
Diabetes mellitus, n (%)	180 (10.1)	262 (11.0)	257 (13.2)	0.009	244 (17.3)	353 (16.9)	352 (17.6)	0.820	424 (13.3)	615 (13.8)	609 (15.4)	0.020
Medications to treat diabetes, n(%)	65 (3.7)	115 (4.9)	98 (5.0)	0.093	97 (6.9)	141 (6.8)	137 (6.9)	0.984	162 (5.1)	256 (5.8)	235 (6.0)	0.259
Hypertension, n (%)	388 (21.9)	620 (26.2)	583 (30.0)	<0.001	477 (33.7)	743 (35.5)	690 (34.6)	0.540	865 (27.1)	1363 (30.5)	1273 (32.3)	<0.001
Dyslipidemia, n(%)	719 (40.6)	921 (39.0)	809 (41.7)	0.194	642 (45.6)	922 (44.2)	845 (42.4)	0.171	1361 (42.8)	1843 (41.4)	1654 (42.0)	0.476
Daily carbohydrate intake (g) ^1^	284.2 ± 107.1	301.7 ± 119.1	319.9 ± 129.4	<0.001	362.3 ± 146.7	362.8 ± 144.6	409.0 ± 175.1	<0.001	318.9 ±132.0	330.4 ± 135.2	365.1 ± 160.6	<0.001
Daily fat intake (g) ^1^	79.5 ± 32.6	80.2 ± 33.7	83.4 ± 37.2	0.001	100.3 ± 44.0	98.6 ± 41.9	105.3 ± 47.3	<0.001	88.7 ± 39.4	88.8 ± 38.9	94.5 ± 44.0	<0.001
Daily protein intake (g) ^1^	117.2 ± 47.0	119.6 ±48.8	125.1 ± 55.6	<0.001	142.4 ± 60.7	142.6 ±61.1	152.2 ± 68.0	<0.001	128.4 ± 55.0	130.4 ± 56.1	138.9 ± 63.7	<0.001
Alcohol intake, n (%)												
Never	249 (14.0)	364 (15.4)	299 (15.4)	0.002	49 (3.5)	89 (4.2)	92 (4.6)	0.002	298 (9.3)	453 (10.2)	391 (9.9)	<0.001
Former user	303 (17.1)	461 (19.5)	422 (21.8)		276 (19.5)	357 (17.1)	431 (21.6)		579 (18.1)	818 (18.3)	853 (21.7)	
Current user	1223 (68.9)	1543 (65.1)	1216 (62.8)		1090 (77.0)	1646 (78.7)	1472 (73.8)		2313 (72.5)	3189 (71.5)	2688 (68.3)	
Smoking status, n (%)												
Never smoker	1071 (60.3)	1573 (66.4)	1218 (62.7)	<0.001	731 (51.7)	1172 (56.0)	1010 (50.5)	<0.001	1802 (56.5)	2745 (61.5)	2228 (56.5)	<0.001
Former smoker	486 (27.4)	512 (21.6)	464 (23.9)		487 (34.4)	649(31.0)	638 (37.9)		973 (30.5)	1161 (26.0)	1102 (28.0)	
Current smoker	219 (12.3)	285 (12.0)	261 (13.4)		197 (13.9)	271 (12.9)	350 (17.5)		416 (13.0)	556 (12.5)	611 (15.5)	

^1.^ Mean and standard deviation. ^2^ BMI was categorized as underweight (<18.5 kg/m^2^), normal weight (18.5–24.9 kg/m^2^), overweight (25–29.9 kg/m^2^) and obesity (≥30 kg/m^2^) ^3.^ Waist circumference was considered as cardiovascular risk as ≥88 cm in women and ≥102 cm in men.

**Table 3 ijerph-20-00806-t003:** Association of physical activity at leisure time and at commuting with diabetes in baseline participants of ELSA-Brasil study (n = 11,797).

			Odds Ratio (95% Confidence Interval)						
	Men			Women			Overall Sample		
	Physically Inactive	Insufficiently Active	Physically Active	Physically Inactive	Insufficiently Active	Physically Active	Physically Inactive	Insufficiently Active	Physically Active
LTPA									
Crude	Reference	0.83 (0.68–1.03)	0.46 (0.39–0.56)	Reference	0.72 (0.54–0.94)	0.63 (0.50–0.78)	Reference	0.84 (0.71–0.98)	0.56 (0.48–0.64)
Model 1	Reference	0.92 (0.74–1.15)	0.60 (0.49–0.72)	Reference	0.79 (0.59–1.06)	0.73 (0.58–0.92)	Reference	0.89 (0.75–1.05)	0.65 (0.56–0.75)
Model 2	Reference	1.04 (0.80–1.36)	0.74 (0.58–0.94)	Reference	0.70 (0.48–1.02)	0.78 (0.59–1.05)	Reference	0.92 (0.74–1.14)	0.76 (0.63–0.91)
CPA									
Crude	Reference	0.97 (0.81–1.16)	1.02 (0.85–1.22)	Reference	1.10 (0.90–1.35)	1.35 (1.10–1.65)	Reference	1.04 (0.91–1.19)	1.19 (1.04–1.36)
Model 1	Reference	0.87 (0.72–1.05)	0.80 (0.66–0.97)	Reference	0.93 (0.75–1.14)	1.03 (0.84–1.28)	Reference	0.90 (0.78–1.03)	0.90 (0.78–1.04)
Model 2	Reference	0.91 (0.72–1.15)	0.92 (0.73–1.17)	Reference	0.87 (0.67–1.13)	1.02 (0.79–1.32)	Reference	0.89 (0.75–1.05)	0.97 (0.81–1.15)

LTPA: Leisure-time physical activity; CPA: Commuting physical activity. Model 1: adjustment by age, educational level and race. Model 2: adjustment by age, educational level, race, waist circumference, hypertension, smoking, alcohol intake, use of medications to treat diabetes and by commuting or leisure-time physical activity (according to the main independent variable). Bold values were statistically significant.

**Table 4 ijerph-20-00806-t004:** Association of physical activity at leisure time and at commuting with pre-diabetes in baseline participants of ELSA-Brasil study (n = 11,797).

			Odds Ratio (95% Confidence Interval)						
	Men			Women			Overall Sample		
	Physically Inactive	Insufficiently Active	Physically Active	Physically Inactive	Insufficiently Active	Physically Active	Physically Inactive	Insufficiently Active	Physically Active
LTPA									
Crude	Reference	0.96 (0.81–1.14)	0.75 (0.65–0.85)	Reference	0.74 (0.62–0.88)	0.67 (0.59–0.77)	Reference	0.92 (0.81–1.03)	0.80 (0.73–0.88)
Model 1	Reference	0.98 (0.82–1.17)	0.84 (0.73–0.96)	Reference	0.75 (0.62–0.90)	0.72 (0.62–0.83)	Reference	0.85 (0.75–0.97)	0.78 (0.70–0.86)
Model 2	Reference	1.04 (0.86–1.25)	0.93 (0.81–1.07)	Reference	0.79 (0.65–0.95)	0.78 (0.67–0.90)	Reference	0.90 (0.79–1.03)	0.86 (0.77–0.95)
CPA									
Crude	Reference	0.97 (0.84–1.13)	0.94 (0.81–1.09)	Reference	1.02 (0.90–1.16)	1.29 (1.13–1.48)	Reference	1.03 (0.93–1.13)	1.17 (1.06–1.29)
Model 1	Reference	0.94 (0.81–1.10)	0.84 (0.72–0.98)	Reference	0.93 (0.81–1.07)	1.10 (0.95–1.27)	Reference	0.94 (0.85–1.04)	0.97 (0.87–1.08)
Model 2	Reference	0.96 (0.82–1.12)	0.88 (0.75–1.04)	Reference	0.98 (0.85–1.12)	1.14 (0.98–1.32)	Reference	0.98 (0.88–1.08)	1.01 (0.91–1.13)

LTPA: Leisure-time physical activity; CPA: Commuting physical activity. Model 1: adjustment by age, educational level and race. Model 2: adjustment by age, educational level, race, waist circumference, hypertension, smoking, alcohol intake and by commuting or leisure-time physical activity (according to the main independent variable). Bold values were statistically significant.

## Data Availability

The data presented in this study are available upon reasonable request from the corresponding author. The data are not publicly available due to ethical commitments for sensitive patient information.

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
