# Peer review of "Association between Leisure-Time and Commute Physical Activity and Pre-Diabetes and Diabetes in the Brazilian Longitudinal Study of Adult Health (ELSA-Brasil)"

_ijerph, 2023, doi:10.3390/ijerph20010806_

Round 1
Reviewer 1 Report
This cross-sectional study aims to analyze the relationship between 19 leisure-time physical activity (LTPA) and commuting (CPA) domains with diabetes and pre-diabetes in the ELSA-Brazil study. This investigation tries to explain positively when looking at the issue of public health and strategies to improve daily physical activity against diabetes. I think some things need to be clarified for publication, which will help in the overall interpretation and understanding of the results before publication within IJERPH.
Comment 1: The regular intake of medication significantly affects the concentrations of blood biomarkers. Did participants take any medication?
Comment 2: Authors should identify the prevalence of the type of medication (ie: Metformin,...)
Comment 3: Please add medication as a covariate in the analysis and rewrite in light of the new analyses.
Author Response
Reviewer 1
Comment 1:
“The regular intake of medication significantly affects the concentrations of blood biomarkers. Did participants take any medication?
- Answer to comment 1:
We have information about use of medication to treat diabetes in the sample. We added at:
Methods section, page 3, Covariates, lines 5-6:
“…and use of medications to treat diabetes were measured through a structured questionnaire….”
Methods section at page 4 lines 4 and 5:
“ …. (age, education, race, waist circumference, hypertension, smoking, alcohol intake and use of medications to treat diabetes). LTPA models were adjusted by CPA and CPA…”
At Results section at page 4 lines 3-10:
“…46.0% dyslipidemia. Frequency of dyslipidemia was higher in participants with diabetes (51.5%), compared to people without diabetes (40.5%). Global, 50.1% reported pre-diabetes, 14.1% diabetes and 7.8% used at least one medication to treat diabetes. Among participants under pharmacological treatment, 90.4% used metformin, 18.3 glibenclamide, 10.3% glimepiride, 7.4% glicazide, 3% vildagliptin, 2.2% sitagliptin and 1.6% acarbose for diabetes management. Another medications used to diabetes were chlorpropamide, exenatide, pioglitazone, repaglinde, rosiglitazone and saxagliptin with frequencies of use less than 1%.”
We also included information about use of medications to treat diabetes in Tables 1 and 2.
In this new version of the article, new table 3 includes the adjustment for use of medication to treat diabetes. After the inclusion of use of medication to treat diabetes, our findings did not materially change. New Table 3 is at page 10.
Comment 2:
“Authors should identify the prevalence of the type of medication (ie: Metformin,...)”
- Answer to comment 2:
We have information detailed about use of medication to treat diabetes. As we answer in the previous comments, this information is now added to Methods (page 3, lines 5-8), Results (page 4, lines 3-10), and Table 1 and Table 2.
New table 3 including the adjustment for use of drugs to treat diabetes is at page 10.
Comment 3:
“Please add medication as a covariate in the analysis and rewrite in light of the new analyses.”
- Answer to comment 3:
As we explained in the answer to comments 1 and 2, the multivariable models in this new version of the study included an adjustment for medications to treat diabetes.
New Table 3 is at page 10.
Reviewer 2 Report
Overall this paper has been written so nicely, intensively analyzed, and looks great for publication but the data is too old to be considered for publication. A small mistake in line 97 ... and (3) physically inactive; referencing style in lines 239, 240, and 251. Many references are also around 10 years or longer.
Author Response
Reviewer 2
Comment 1:
“Overall this paper has been written so nicely, intensively analyzed, and looks great for publication but the data is too old to be considered for publication.”
- Answer to comment 1:
This study uses data from the ELSA-Brasil baseline that occurred from 2008-2010. The number of studies that evaluated the association between practice of physical activity at leisure and at commute and its relationship to diabetes in the same analysis is very scarce. Therefore, it is possible that these results may help to highlight the possible different impact of practice of physical activity at leisure or at and commute over diabetes. Our results, if confirmed in new analyses, may help to change guidelines or public policies related to the effect of practice of physical activity at commute.
Comment 2:
- “A small mistake in line 97”
- Answer to comment
The mistake was corrected.
- ... and (3) physically inactive; referencing style in lines 239, 240, and 251.”
- Answer to comment 2:
Reference style was corrected.
Comment 3:
“ Many references are also around 10 years or longer.”
- Answer to comment 3:
We changed all possible references. We cannot change some old references that have information about previous published articles that evaluate practice of physical activity at leisure and commute and their association with diabetes, since they have important findings related to the objective of the present analysis.
We added 7 new references:
New Reference 19:
Molina, M.C.; Benseñor, I.M.; Cardoso, L.O.; Velasquez-Melendez, G.; Drehmer, M.; Pereira, TSS.; De Faria, C.P.; Melere, C.; Manato, L.; Gomes, A.L.C.; Da Fonseca, M.J.M.; Sichieri, R. Reproducibility and relative validity of the Food Frequency Questionnaire used in the ELSA-Brasil. Cad Saude Publica 2013, 29, 379–389.
New Reference 25:
Grøntved, A.; Pan, A.; Mekary, R.A.; Stampfer, M.; Willett, W.C.; Manson, J.E.; Hu, F.B. Muscle-strengthening and conditioning activities and risk of type 2 diabetes: A prospective study in two cohorts of US women. Plos Medicine 2014, 11, e1001587.
New reference 27:
Imierska, M.; Kurianiuk, A.; Błachnio-Zabielska, A. The influence of physical activity on the bioactive lipids metabolism in obesity-induced muscle insulin resistance. Biomolecules 2020, 10, 1-20.
New reference 28:
Janochova, K.; Haluzik, M.; Buzga, M. Visceral fat and insulin resistance – what we know?. Biomed Pap Med Fac Univ Palacky Olomouc Czech Repub 2019, 163, 19-27.
New reference 29:
Sabag, A.; Way, K.L.; Keating, S.E.; Sultana, R.N.; O’Connor, H.T.; Baker, M.K.; Chuter, V.H.; George, J.; Johnson, N.A. Exercise and ectopic fat in type 2 diabetes: A systematic review and meta-analysis. Diabetes & Metabolism 2017, 43, 195–210
New reference 30:
Mann, S.; Beedie, C.; Jimenez, A. Differential effects of aerobic exercise, resistance training and combined exercise modalities on cholesterol and the lipid profile: Review, synthesis and recommendations. Sports Med 2014, 44, 211–221.
New reference 31:
Wang, Y.; Xu, D. Effects of aerobic exercise on lipids and lipoproteins. Lipids in Health and Disease 2017, 16, 1-8.
Reviewer 3 Report
Individuals with higher frequency of diabetes or pre-diabetes have any background of metabolic history in the family? Also, how does their lipid profile looks like?
This is just an observation, no data required.
As a suggestion, you can include a little paragraph with relation between PA and dyslipidemia, an important factor that can precede or follow diabetes installing. Subjected to PA, lipid metabolism can influence sugar regulation and future metabolic behavior. A section of their food diet could be also introduce in your tables. Again, just as suggestion.
Author Response
Comment 1:
“Individuals with higher frequency of diabetes or pre-diabetes have any background of metabolic history in the family? Also, how does their lipid profile looks like?”
- Answer to comment 1:
We have information about dyslipidemia in the sample.
We added at Methods section, page 3, Covariates, lines 14-16
“…dyslipidemia was defined as triglycerides ≥150 mg/dL or LDL-cholesterol ≥130 mg/dL or HDL-cholesterol <40 mg/dL in men and <50 mg/dL in women or taking lipid-lowering medication….”
We added at Results section at page 4:
“… 46.0% of dyslipidemia….”
We added ad Results section at page 4:
“Frequency of dyslpidemia was higher in participants with diabetes (51.5%), compared to people without diabetes (40.5%).”
We also included information about dyslipidemia in Tables 1 and 2 (pages 5-8).
Comment 2:
“This is just an observation, no data required.
As a suggestion, you can include a little paragraph with relation between PA and dyslipidemia, an important factor that can precede or follow diabetes installing. Subjected to PA, lipid metabolism can influence sugar regulation and future metabolic behavior. A section of their food diet could be also introduce in your tables. Again, just as suggestion”.
- Answer to comment 2:
We introduced a new paragraph at the Discussion section as you suggested. The new paragraph at the page 14 at the Discussion section, third paragraph:
“The inverse association observed between PA and diabetes can be explained through weight control and improvements in the lipid profile. Chronic elevation of plasmatic lipids contributes to their accumulation in body tissues, such as skeletal muscle and liver, heart and pancreas; in turn, it can lead to insulin resistance [27-29]. There are evidence suggesting that the regular practice of PA may be associated not only with a better weight control, but also with an enhancement in the lipid profile [4,30,31]; additionally, PA can contribute to decreased fat mass, particularly visceral fat[29], which is more related to insulin resistance than subcutaneous fat[28].”
We also have information about diet from a food frequency questionnaire and the information about intake of carbohydrate, fat and protein as part of new Tables 1 and 2.
We added at Methods section at page 3, Covariates, pages 6-8:
“…Macronutrients daily intake was calculated based on information from a food frequency questionnaire validated for the study[19]…”
We also included this information in Tables 1 and 2 (pages 5-8).